# Development of a Deep Learning Model for Malignant Small Bowel Tumors Survival: A SEER-Based Study

**DOI:** 10.3390/diagnostics12051247

**Published:** 2022-05-17

**Authors:** Minyue Yin, Jiaxi Lin, Lu Liu, Jingwen Gao, Wei Xu, Chenyan Yu, Shuting Qu, Xiaolin Liu, Lijuan Qian, Chunfang Xu, Jinzhou Zhu

**Affiliations:** 1Department of Gastroenterology, The First Affiliated Hospital of Soochow University, Suzhou 215006, China; myyin4984@stu.suda.edu.cn (M.Y.); jxlin@stu.suda.edu.cn (J.L.); 20205232070@stu.suda.edu.cn (L.L.); 20204232009@stu.suda.edu.cn (J.G.); xw222300@163.com (W.X.); 20195232011@stu.suda.edu.cn (C.Y.); 20215232078@stu.suda.edu.cn (S.Q.); lxl55@foxmail.com (X.L.); 20204132060@stu.suda.edu.cn (L.Q.); 2Suzhou Clinical Center of Digestive Diseases, Suzhou 215006, China

**Keywords:** DeepSurv, Cox proportional hazards, survival analysis, small bowel tumors, SEER database

## Abstract

**Background** This study aims to explore a deep learning (DL) algorithm for developing a prognostic model and perform survival analyses in SBT patients. **Methods** The demographic and clinical features of patients with SBTs were extracted from the Surveillance, Epidemiology and End Results (SEER) database. We randomly split the samples into the training set and the validation set at 7:3. Cox proportional hazards (Cox-PH) analysis and the DeepSurv algorithm were used to develop models. The performance of the Cox-PH and DeepSurv models was evaluated using receiver operating characteristic curves, calibration curves, C-statistics and decision-curve analysis (DCA). A Kaplan–Meier (K–M) survival analysis was performed for further explanation on prognostic effect of the Cox-PH model. **Results** The multivariate analysis demonstrated that seven variables were associated with cancer-specific survival (CSS) (all *p* < 0.05). The DeepSurv model showed better performance than the Cox-PH model (C-index: 0.871 vs. 0.866). The calibration curves and DCA revealed that the two models had good discrimination and calibration. Moreover, patients with ileac malignancy and N2 stage disease were not responding to surgery according to the K–M analysis. **Conclusions** This study reported a DeepSurv model that performed well in CSS in SBT patients. It might offer insights into future research to explore more DL algorithms in cohort studies.

## 1. Background

Previous studies indicate that small bowel tumors (SBTs) are rare, with an estimated incidence rate of 2.6 per 100,000 for males and 2.0 per 100,000 for females in 2019 [1]. Despite the fact that the small intestine takes up the majority of the whole gastrointestinal tract, SBTs only account for 1–2% of all gastrointestinal tumors [2]. Nevertheless, SBTs have been on the rise for decades, with an average annual growth rate of 1.8%, owing to the remarkable progress of endoscopy and imaging [1]. Malignancy is more frequent in SBTs, with the most common histological diagnoses being adenocarcinoma, neuroendocrine tumors, gastrointestinal stromal tumors (GISTs) and lymphoma [3]. SBTs are characterized by a lengthy silence-period, nonspecific symptoms and a high degree of malignancy [1,4]. Therefore, studies that focus on the prognosis of SBTs are largely restricted.

Artificial intelligence (AI), consisting of machine learning (ML) and deep learning (DL), has lately been applied to the field of medicine [5,6,7,8,9]. Using an ensemble of ML models, Levi et al. [8] developed a prediction model for detecting patients with gastrointestinal bleeding who required transfusion. Skrede et al. [5] trained a DL model, based on ten-layer convolutional neural networks (CNNs), for risk stratification to help decision making in the therapy of colorectal cancer. Therefore, AI-aided systems are useful in various clinical settings, e.g., risk stratification, personalized treatments and disease classification based on endoscopic images [10,11,12]. As for AI application in survival analysis, Katzman et al. [13] first proposed a novel technology, based on deep neural networks, for survival analysis in 2018. They proved that the use of DeepSurv allowed for higher performance of prognostic prediction and effective treatment recommendations, which was superior to the state-of-the-art survival methods such as Cox-PH and random survival forests. Since then, DeepSurv has been used in a variety of fields [14,15,16], e.g., the prognosis of lung cancer [17], the therapeutic efficacy of chemoradiation in head and neck cancer [18] and the detection of cognitive decline [19], which all showed better performance than traditional Cox-PH models.

To date, several survival models have been proposed for predicting overall survival (OS) and cancer-specific survival (CSS) in SBTs [20,21,22,23,24]. However, these models all used traditional Cox proportional hazards methods (Cox-PH), which were time-consuming due to manual feature selection. Furthermore, the Cox-PH is a semiparametric model that explores the linear correlation between covariates, even though relations among covariates and survival outcomes are usually complex and nonlinear. Therefore, our study aims to evaluate the feasibility of DL models for survival analysis in malignant SBTs, based on the Surveillance, Epidemiology and End Results (SEER) dataset [13].

## 2. Materials and Methods

### 2.1. Cohorts Dataset

This study was based on the SEER database that provides information on cancer statistics, such as demographic and clinical data, in an effort to reduce the cancer burden [10,11,13,14]. SBTs were diagnosed using the third edition of the International Classification of Disease for Oncology (ICD-O-3), with site codes of C17.0–17.9. For cases diagnosed in 2018 and later, the SEER program applied Extent of Disease (EOD), a set of three items including primary tumor, lymph nodes and metastasis. According to the aforementioned criteria, cases diagnosed with SBTs in 2018 were included in this study. All patients were selected from the “SEER Incidence Dataset”, which collected cancer incidence data from population-based cancer registries covering nearly 47.9% of the U.S. population, using SEER*Stat software (version 3.8.9.2). Moreover, the following are the exclusion criteria: (1) patients under the age of 18; (2) patients whose survival duration was unclear; (3) patients with incomplete demographic data. We collected variables from the SEER database, including age, sex, race, income, region, T stage, N stage, M stage, tumor size, primary tumor site, histological type and the number of lymph nodes removed. Our primary endpoint was CSS, the time interval between the first diagnosis and death from SBTs. Figure 1 shows the flowchart of this study.

### 2.2. Data Preprocessing

A total of 1987 patients were included in our study and were randomly split into the training set (*n* = 1398) and validation set (*n* = 589) at 7:3. Missing data, assumed to be missing at random, were multiple imputed using a random forest algorithm from the ‘mice’ package in R software [25]. The imputed variables were tumor size (21.67% in the training set and 21.05% in the validation set) and the number of lymph nodes removed (4.94% in the training set and 2.89% in the validation set).

### 2.3. Development of the Cox-PH Model

Univariate and multivariate Cox regression analyses were used to select important features and develop a prognostic model. The variables with a two-sided *p*-value of less than 0.05, determined in the univariate analysis, were integrated into the multivariate analysis. The performances of the 3-, 6- and 9-month survival models were evaluated with time-dependent receiver operating characteristic (ROC) curve, calibration curve, Harrell C-statistics and decision-curve analysis (DCA) based on the training and validation sets.

### 2.4. Development of the DeepSurv Model

DeepSurv is a feed-forward neural-network-based extension of the Cox proportional hazards model, consisting of an input layer, an output layer and multiple hidden layers, which can predict the survival outcomes. Patients’ baseline data are fed into the network as an input layer. A fully connected layer of nodes is followed by a dropout layer in the hidden layer. The output layer generates the probability of each patient’s survival event estimated using the DeepSurv model. To begin, we used the training cohort, which was randomly split into the internal training set and internal validation set at 8:2, to develop the DL model of a four-layer neural network, and the validation cohort to evaluate the efficacy of this model. The process of training involved making real-time adjustments to the hyperparameters, which included the input standardization, the Rectified Linear Units (ReLU) activation function, the Adaptive Moment Estimate (Adam) optimizer and the learning rate scheduling [13]. Early stopping regularization was used to prevent model overtraining and avoid overfitting. The performance of the model was evaluated using the Harrell C-index, ROC curve, calibration curve and DCA based on the validation set.

### 2.5. Statistical Analysis

Continuous variables were presented as the mean ± standard deviation (SD) if normally distributed or as the median if not. Categorical variables were described as frequencies. Baseline characteristics were compared using Student’s t-test or nonparametric tests for continuous variables and Chi-square tests for categorical variables. Furthermore, Kaplan–Meier (K–M) survival analysis was applied to evaluate the prognostic effect of those independent factors determined with multivariate Cox regression analysis using the log-rank test. Statistical analysis was performed using the ‘MASS’ package of R software (version 4.0) and the ‘pycox’ package of Python (version 3.9). A two-side *p* < 0.05 was considered statistically significant.

## 3. Results

### 3.1. Patient Characteristics

Our study involved a total of 1987 patients with malignant SBTs, with more males (53.6%) than females (46.4%). A majority of patients with SBTs were between the age of 40 and 80, were white (74.1%), and lived in the metropolis (89.1%) with a comfortable household income. Carcinoid (50.4%) was the most common histological type, followed by adenocarcinoma (30.9%) and stromal sarcoma (11.0%). Primary tumor sites were located in the duodenum (35.1%), ileum (33.3%), jejunum (11.0%) and overlapping sites (0.8%). Based on the updated American Joint Committee on Cancer (AJCC) 8th edition staging criteria, patients were classified as stage I (24.8%), II (15.5%), III (25.2%) and IV (24.7%). More details are listed in Table 1.

### 3.2. Performance of the Cox-PH Model

The dataset was randomly split into the training set (*n* = 1398) and the validation set (*n* = 589) at 7:3. Multivariate Cox regression analysis was performed to determine independent prognostic factors for CSS, including age, race, primary tumor site, histological type, N stage, surgery and the number of retrieval lymph nodes, as shown in Figure 2. The results showed that patients who were Asian, had tumors located in the ileum and had N2 stage disease had poor prognosis (*p* < 0.05). Additionally, histological types had the greatest impact on prognosis, of which adenocarcinoma and others (neuroendocrine carcinoma, squamous cell carcinoma, signet ring carcinoma, etc.) accounted for a large proportion (*p* < 0.001).

The C-indices of the Cox-PH model were 0.878 and 0.866, respectively, based on the training and validation set, as shown in Appendix A. Furthermore, ROC curves (Figure 3A) were constructed to evaluate the discrimination capability of this model based on the validation set, with the areas under the ROC curve (AUCs) of 0.874, 0.922 and 0.908 for 3-, 6- and 9-month survival probability, respectively. Figure 4A–C shows that our model had minor deviations from the reference line in the calibration curve for 3-, 6- and 9-month CSS in the validation set, with the Brier scores of 0.282, 0.265 and 0.265, respectively. Furthermore, the C-index, AUCs and Brier scores based on the training set are demonstrated in Table 2. Additionally, DCA curves for this Cox-PH model are presented in Figure 5A–C for 3-, 6- and 9-month CSS, demonstrating that our model could help patients obtain a net benefit of 8–10%.

### 3.3. Performance of the DeepSurv Model

The aforementioned training set (*n* = 1398) was randomly split into the “internal training” set (*n* = 1129) and the “internal validation” set (*n* = 269) at 8:2 for the best set of hyperparameters. The validation set (*n* = 589) was regarded as the “testing” set for evaluating the performance of the final DeepSurv model. We tuned DeepSurv’s hyperparameters by referring to previous studies [7,13], as described in Appendix A. With the best performing hyperparameters, four layers (nodes: 32, 64, 128, 256), a dropout rate of 0.2 and a learning rate of 0.020092, the performance of the final model was evaluated with the validation set (*n* = 589). The C-index was 0.871, while the AUCs were 0.878, 0.891 and 0.891 for 3-, 6- and 9-month CSS, respectively, which were inferior to those of the Cox-PH model, as shown in Table 2 and Figure 3B. Furthermore, the calibration and prediction curves of the DeepSurv model for the 3-, 6- and 9-month CSS showed great correlations in the validation cohort, with Brier scores of 0.058, 0.070 and 0.080, respectively, indicating a high degree of reliability (Figure 4D–F). DCA indicated that this model had similar net benefits as the Cox-PH model, demonstrating both of them possess good clinical practice in predicting CSS in patients with SBTs (Figure 5D–F).

### 3.4. K–M Survival Analysis Based on the COX-PH Model

Survival analysis showed that patients who were old and had adenocarcinoma had poorer prognosis (*p* < 0.0001), while patients who had a tumor located in the ileum had relatively good prognosis (*p* < 0.001), as shown in Appendix A. Furthermore, we conducted a sub-group analysis in patients who received surgery. It demonstrated that surgery could significantly improve the CSS whatever the histological type was (Appendix A, *p* < 0.0001). For patients who had a tumor located in the ileum and had N2 stage disease, surgery showed no significant efficacy in improving prognosis (Appendix A, *p* > 0.05).

## 4. Discussion

Due to the relatively low prevalence, studies concerned with the management of SBTs are lacking. Therefore, effective methods for the risk stratification and personalized therapy of SBTs are essential for improving their survival rates.

Currently, it is well known that the AJCC stage system is applied to predict the survival of patients with SBTs. However, this system only takes TNM stage into consideration for predicting cancer risk, neglecting other significant clinical features and demographic information. To improve predictive accuracy, recent studies have constructed various prognostic models for small bowel adenocarcinoma and neuroendocrine carcinoma, using Cox regression analysis [20,21,22,24,26]. Wang et al. [20] built and verified nomograms to predict OS and CSS for small intestine adenocarcinoma (SIA), with higher sensitivity and specificity than the AJCC stage system. That study highlighted ten variables, including demographic data (age and marital status) and therapeutic regimen (surgery and chemotherapy). Zheng et al. [21] also fitted a Cox-PH model for predicting the CSS of SIA, with the C-index of 0.728 based on the internal validation. Cox regression analysis achieves high accuracy and has become a popular method for prognostic predictions. Nevertheless, it is impractical to fit survival models based on linear relation, which requires more complex training methods, e.g., ML algorithms or even deep neural networks.

The remarkable progress in AI facilitates the development of complex nonlinear models based on big clinical data in survival analysis. Adeoye et al. [7] discovered that the DeepSurv model had a higher C-index (0.95 vs. 0.83) for assessing the malignant-transformation risk in patients with oral leukoplakia and lichenoid lesions than traditional Cox regression analysis. As mentioned in the ‘Introduction’ section, it is indeed practicable to apply the DeepSurv algorithm to fitting prognostic models. However, to our knowledge, relative research using DL algorithms for prognostic prediction in SBTs are limited. Therefore, this study presented the development of a DeepSurv model for predicting the CSS of patients with SBTs and compared the performance of the Cox-PH and DeepSurv models.

Our study concluded that the DeepSurv model performed better in predicting survival for patients with SBTs than the Cox-PH model, based on their C-indices. Additionally, calibration curves demonstrated that the estimated risk when using DeepSurv was closer to the observed risk than when using Cox-PH analysis, with lower Brier scores. DCA showed that the two models obtained similar performance in clinical practice, with 8–10% net benefits. As a result, the predictions estimated using this DeepSurv model were proven to have good discrimination and calibration. The advantages of the DeepSurv model, we believe, attribute to the automated features extraction of inputs and the analytic ability of complex big data [17,27,28].

In terms of K–M analysis, our study demonstrated that surgery was a good choice for improving prognosis in patients with SBTs. However, in the sub-group analysis, there was some controversy over surgery. Firstly, patients with ileac tumors were proven to have better CSS compared to those with duodenal or jejunal cancer in our study. Similarly, Nicholl et al. [29] drew the same conclusion from analyzing 1444 patients with small intestine adenocarcinoma, whereas for further analysis, we found that patients with ileac malignancy were not responding to surgery according to the K–M survival analysis. Secondly, surgery had an adverse effect on patients with N2 stage disease, who were classified into stage IIIB-IV, in our study. Many research studies have revealed that palliative chemotherapy could prolong life for stage IV patients, no matter whether surgery was performed or not [30,31]. Therefore, chemotherapy might become another alternative for patients with an advanced stage of SBT. More research is needed to further explore more effective and precise treatment for different sub-group SBT patients.

There are some limitations in our study. To begin with, this was a retrospective study based on the SEER dataset, with potential heterogeneity of the enrolled patients. More external validations are needed in future research. Secondly, more risk factors could be included in future research to better guide the therapeutic regimen for SBT patients.

## 5. Conclusions

In conclusion, our study found that the DeepSurv model performed better than the traditional Cox-PH model in predicting the survival of patients with SBTs. The DL model exhibited feasibility and promise in clinical practices, which might offer insights for future research to explore more DL algorithms in prognostic analysis.

## Figures and Tables

**Figure 1 diagnostics-12-01247-f001:**
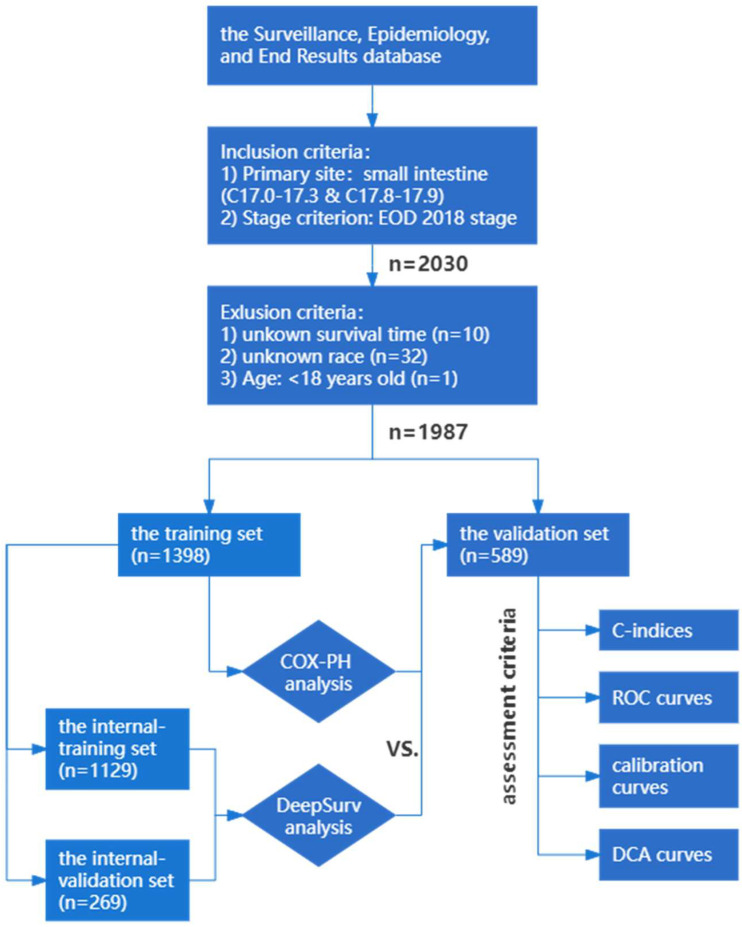
Flowchart of this study.

**Figure 2 diagnostics-12-01247-f002:**
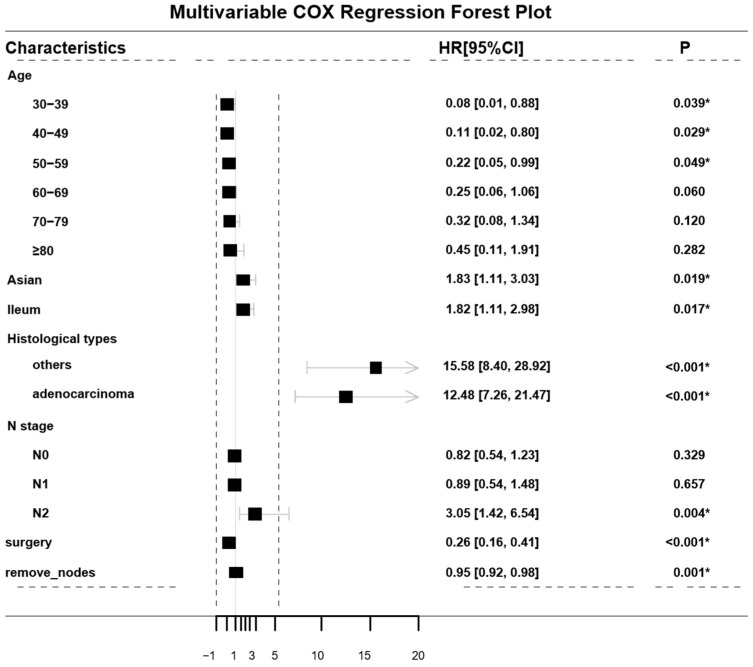
Multivariate Cox regression analysis. * *p* < 0.05 was considered statistically significant.

**Figure 3 diagnostics-12-01247-f003:**
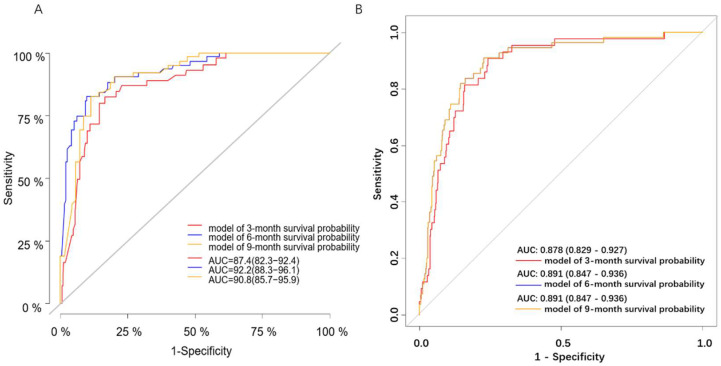
ROC curves of the Cox-PH (**A**) and DeepSurv (**B**) models.

**Figure 4 diagnostics-12-01247-f004:**
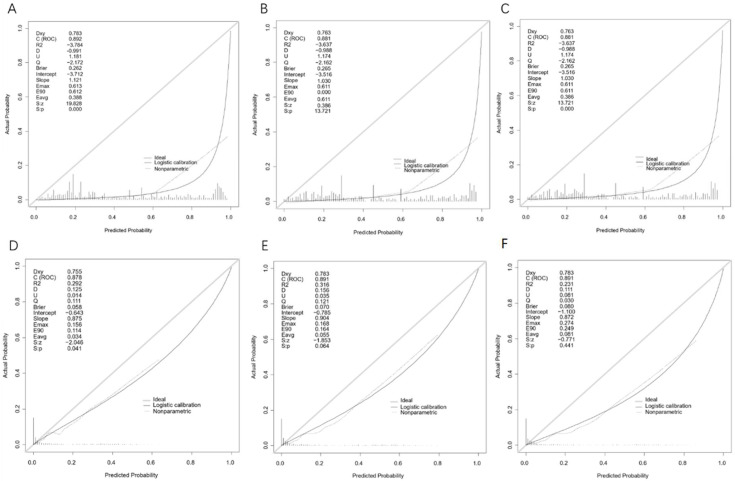
Calibration curves of the Cox-PH (**A**–**C**) and DeepSurv (**D**–**F**) models for 3-, 6- and 9-month cancer-specific survival.

**Figure 5 diagnostics-12-01247-f005:**
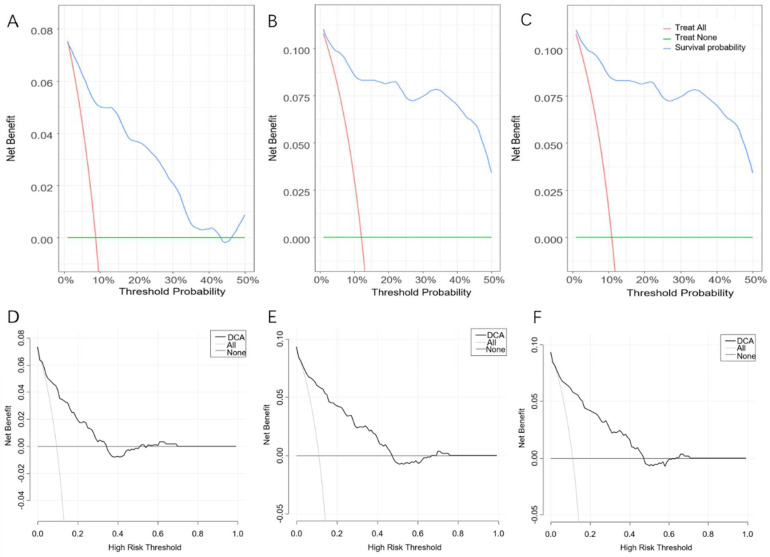
Decision-curve analysis of the Cox-PH (**A**–**C**) and DeepSurv (**D**–**F**) models for 3-, 6- and 9-month cancer-specific survival.

**Table 1 diagnostics-12-01247-t001:** Characteristics of patients included in our study.

	The Training Set (*n* = 1398)			The Validation Set (*n* = 589)		
Characteristics	Non-CSD (*n* = 1238)	CSD (*n* = 160)	*p*	Non-CSD (*n* = 534)	CSD (*n* = 55)	*p*
**Sex (%)**			0.404			0.320
**Female**	566 (45.7)	67 (41.9)		258 (48.3)	31 (56.4)	
**Male**	672 (54.3)	93 (58.1)		276 (51.7)	24 (43.6)	
**Age (years, %)**			<0.001			<0.001
**<30**	7 (0.6)	2 (1.2)		4 (0.7)	0 (0.0)	
**30–39**	40 (3.2)	1 (0.6)		20 (3.7)	1 (1.8)	
**40–49**	100 (8.1)	2 (1.2)		42 (7.9)	1 (1.8)	
**50–59**	249 (20.1)	16 (10.0)		119 (22.3)	10 (18.2)	
**60–69**	382 (30.9)	39 (24.4)		142 (26.6)	8 (14.5)	
**70–79**	308 (24.9)	53 (33.1)		143 (26.8)	14 (25.5)	
**≥80**	152 (12.3)	47 (29.4)		64 (12.0)	21 (38.2)	
**Race (%)**			0.056			0.199
**Black**	246 (19.9)	29 (18.1)		89 (16.7)	11 (20.0)	
**White**	914 (73.8)	112 (70.0)		408 (76.4)	39 (70.9)	
**Asian**	70 (5.7)	18 (11.2)		33 (6.2)	3 (5.5)	
**Indian**	8 (0.6)	1 (0.6)		4 (0.7)	2 (3.6)	
**Median household Income ($, %)**			0.732			0.984
**(−∞, 35,000)**	19 (1.5)	1 (0.6)		10 (1.9)	1 (1.8)	
**[35,000, 55,000)**	271 (21.9)	38 (23.8)		106 (19.9)	12 (21.8)	
**[55,000, 75,000)**	539 (43.5)	66 (41.2)		222 (41.6)	23 (41.8)	
**[75,000, +∞)**	409 (33.0)	55 (34.4)		196 (36.7)	19 (34.5)	
**Region**			0.386			0.743
**non-metropolis**	141 (11.4)	14 (8.8)		55 (10.3)	7 (12.7)	
**metropolis**	1097 (88.6)	146 (91.2)		479 (89.7)	48 (87.3)	
**Primary tumor site (%)**			<0.001			<0.001
**NOS**	131 (10.6)	18 (11.2)		51 (9.6)	5 (9.1)	
**Duodenum**	537 (43.4)	102 (63.7)		210 (39.3)	38 (69.1)	
**Jejunum**	145 (11.7)	15 (9.4)		57 (10.7)	2 (3.6)	
**Ileum**	419 (33.8)	23 (14.4)		210 (39.3)	9 (16.4)	
**Overlapping**	6 (0.5)	2 (1.2)		6 (1.1)	1 (1.8)	
**Histological type (%)**			<0.001			<0.001
**others**	72 (5.8)	31 (19.4)		40 (7.5)	10 (18.2)	
**Adenocarcinoma**	320 (25.8)	111 (69.4)		143 (26.8)	39 (70.9)	
**Carcinoid**	685 (55.3)	16 (10.0)		295 (55.2)	6 (10.9)	
**Stromal sarcoma**	161 (13.0)	2 (1.2)		56 (10.5)	0 (0.0)	
**T stage (%)**			<0.001			0.001
**Tx**	164 (13.2)	68 (42.5)		77 (14.4)	23 (41.8)	
**T0**	3 (0.2)	2 (1.2)		0	0	
**Tis**	0 (0.0)	1 (0.6)		0	0	
**T1**	298 (24.1)	22 (13.8)		112 (21.0)	9 (16.4)	
**T2**	176 (14.2)	5 (3.1)		84 (15.7)	1 (1.8)	
**T3**	340 (27.5)	17 (10.6)		146 (27.3)	5 (9.1)	
**T4**	257 (20.8)	45 (28.1)		115 (21.5)	17 (30.9)	
**N stage (%)**			0.731			0.556
**Nx**	109 (8.8)	37 (23.1)		53 (9.9)	12 (21.8)	
**N0**	687 (55.5)	79 (49.4)		260 (48.7)	27 (49.1)	
**N1**	303 (24.5)	29 (18.1)		150 (28.1)	11 (20.0)	
**N2**	139 (11.2)	15 (9.4)		71 (13.3)	5 (9.1)	
**M stage (%)**			<0.001			<0.001
**M1**	983 (79.4)	78 (48.8)		408 (76.4)	27 (49.1)	
**M2**	255 (20.6)	82 (51.2)		126 (23.6)	28 (50.9)	
**Stage (%)**			<0.001			<0.001
**NA**	117 (9.5)	27 (16.9)		42 (7.9)	9 (16.4)	
**I**	342 (27.6)	16 (10.0)		132 (24.7)	3 (5.5)	
**II**	207 (16.7)	17 (10.6)		77 (14.4)	6 (10.9)	
**III**	317 (25.6)	18 (11.2)		157 (29.4)	9 (16.4)	
**IV**	255 (20.6)	82 (51.2)		126 (23.6)	28 (50.9)	
**Surgery (%)**			<0.001			<0.001
**No**	288 (23.3)	114 (71.2)		123 (23.0)	40 (72.7)	
**Yes**	950 (76.7)	46 (28.7)		411 (77.0)	15 (27.3)	
**Tumor size** **(mm, median [IQR])**	21.00 [12.00, 42.00]	32.00 [20.00, 55.00]	<0.001	22.00 [12.00, 41.00]	40.00 [25.50, 53.00]	<0.001
**Removed nodes** **(median [IQR])**	3.00 [0.00, 15.00]	0.00 [0.00, 1.00]	<0.001	3.00 [0.00, 15.00]	0.00 [0.00, 0.00]	<0.001
**Lymph node examine (%)**			<0.001			<0.001
**No**	522 (42.2)	115 (71.9)		222 (41.6)	41 (74.5)	
**Yes**	716 (57.8)	45 (28.1)		312 (58.4)	14 (25.5)	

CSD = cancer-specific death; NOS = no other specific; NA = not available. The *p*-values for T-stage and N stage were calculated excluding Tx and Nx.

**Table 2 diagnostics-12-01247-t002:** Performance of the COX-PH and DeepSurv models.

Model	Group	C-index	AUCs		Brier Scores			
			3-Month	6-Month	9-Month	3-Month	6-Month	9-Month
Cox-PH	The training set (*n* = 1398)	0.878	0.898	0.897	0.894	0.262	0.252	0.246
	The validation set (*n* = 589)	0.866	0.874	0.922	0.908	0.282	0.265	0.265
DeepSurv	The internal validation set (*n* = 269)	0.869	0.874	0.871	0.871	0.068	0.079	0.085
	The validation set (*n* = 589)	0.871	0.878	0.891	0.891	0.058	0.070	0.080

## Data Availability

The data presented in this study are openly available in https://seer.cancer.gov/ (accessed on 20 March 2022).

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
