# Peer review of "Development of a Deep Learning Model for Malignant Small Bowel Tumors Survival: A SEER-Based Study"

_diagnostics, 2022, doi:10.3390/diagnostics12051247_

Round 1
Reviewer 1 Report
This manuscript reports a DeepSurv model for small bowel tumors based on a public SEER data and compares its performance with that based on the traditional Cox proportional hazard (COX-PH) model. The authors claim that the DeepSurv model performed well and showed some better characteristics such as DCA and Brier index than COX-PH. Regarding clinical relevance, they mention that ileac malignancy and N2 stage patients do not benefit from surgery based on Kaplan-Meier plots.
The most serious flaw in this work is that they split the dataset into two groups, called “training” and “validation”. In the area of machine learning or deep learning which requires hyperparameter tuning, one should split the dataset into three groups: “training”, “validation”, and “testing”. The “testing” set corresponds to this work’s “validation” set. Given a set of hyperparameters, one trains the model based on the “training” set and evaluates its performance on the “validation” set. Among various sets of hyperparameters, the set showing the best performance in the “validation” set is chosen. This model is finally evaluated in the “testing” set and its performance is reported. This is the typical process in ML/DL to avoid overfitting hyperparameters. Unfortunately, the authors reported the performance of various sets of hyperparameters based on their validation data and have chosen the best one, which was claimed as the final result. This is not an acceptable practice.
Let alone the issue of hyperparameter tuning, there is another issue which should draw attention. DeepSurv not only can perform better than COX-PH in general, but also can recommend personalized treatment options. The original article reporting the DeepSurv system emphasizes this aspect in its title: “DeepSurv: personalized treatment recommender system using a Cox proportional hazards deep neural network”. This feature of DeepSurv was mentioned in this manuscript as well in Discussion line 208. This manuscript also reports a section of treatment recommendation, but I cannot find any evidence that this analysis was done by DeepSurv. It appears that the results were an elaboration of the results from COX-PH through K-M plots. If this part was really based on COX-PH, I wonder why it was reported as one of the major findings, worthy to be mentioned in Abstract, Results, and Discussion. By not providing specific details on this analysis, the readers may be misled to the impression that it is based on DeepSurv, as the title emphasizes that this manuscript is about DeepSurv.
There are numerous minor comments:
- Line 19: “CSS” abbreviation was used in Abstract without full words
- Lines 41-45: A brief review of DL on medicine is too broad and brief. It would be better to focus of DL in survival analysis as it is the theme of this manuscript. I think a paragraph in Discussion (lines 204-216) may be a good introduction.
- In numerous places, “COX regression” was used. It should be replaced to “Cox regression”
- Line 62: SEER is not “open-source”
- Line 104: “cross-validation” was used. Please elaborate on this
- Table A1: It seems that T stage and N stage were significant due to Tx and Nx. They represent missing data and it is awkward that statistical significance was obtained by missing data.
- Table A2: “Katzman et al. [15]” should be “[18]”
- Lines 178 & 233: The authors claim that N2 stage does not benefit from surgery. But it appears N4 has the adverse effect.
- Figure A6 is illegible. Its quality should be improved. “Age” was categorized to 0-6, without definition.
- Line 197: “This study” represents yours. It should be “That study”.
- Lines 253-254 should be removed
Author Response
Major comments:
Point 1:The most serious flaw in this work is that they split the dataset into two groups, called “training” and “validation”. In the area of machine learning or deep learning which requires hyperparameter tuning, one should split the dataset into three groups: “training”, “validation”, and “testing”. The “testing” set corresponds to this work’s “validation” set. Given a set of hyperparameters, one trains the model based on the “training” set and evaluates its performance on the “validation” set. Among various sets of hyperparameters, the set showing the best performance in the “validation” set is chosen. This model is finally evaluated in the “testing” set and its performance is reported. This is the typical process in ML/DL to avoid overfitting hyperparameters. Unfortunately, the authors reported the performance of various sets of hyperparameters based on their validation data and have chosen the best one, which was claimed as the final result. This is not an acceptable practice.
Response 1:Many thanks for your kind reminder and we agree with your point on the typical process in ML/DL. We have revised the fitting process of the DeepSurv model. The steps are as followed: 1). The “training” set (n=1398) was randomly split into the “internal training” set (n=1129) and the “internal validation” set (n=269) at 8:2. The “validation” set (n=589) was regarded as the “testing” set for evaluating the performance of DeepSurv model. 2). The DeepSurv model was retraining based on the “internal training” set and the hyperparameters were adjusted according to the performance on the “internal validation” set. 3). Among various sets of hyperparameters, the set showing the best performance in the “internal validation” set was chosen. Finally, the DeepSurv model was evaluated in the “validation” set, showing the C-index of 0.871. Furthermore, the performance of DeepSurv model, including AUCs, Brier scores and C-index, and plots of ROC curves, calibration curves and DCA curves all get changed.
- Change1:
2.Materials and Methods
2.4. Development of the DeepSurv model
“Line 7-9: To begin, we used the training cohort, which was randomly split into the internal-training set and internal-validation set at 8:2, to develop the DL model of a four-layer neural network, and the validation cohort to evaluate the efficacy of this model. “
- Change2:
3.Results
3.3. Performance of the DeepSurv model
“Line 1-12: The aforementioned training set (n=1398) was randomly split into the “internal training” set (n=1129) and the “internal validation” set (n=269) at 8:2 for the best set of hyperparameters. The validation set (n=589) was regarded as the “testing” set for evaluating the performance of the final DeepSurv model. The highest C-index of the DeepSurv model was 0.871, with four layers (nodes: 32, 64, 128, 256), dropout rate of 0.2 and learning rate of 0.020092. We tuned DeepSurv’s hyperparameters by referring to previous studies [7, 13], as described in Table A2. Based on the validation set, the AUCs were 0.878, 0.891 and 0.891 for 3-, 6- and 9-month CSS, respectively, which were inferior to those of the Cox-PH model, as shown in Fig. A3B. Furthermore, the calibration and prediction curves of the DeepSurv model for the 3-, 6- and 9-month CSS showed great correlations in the validation cohort, with brier scores of 0.058, 0.070 and 0.080, respectively, indicating a high degree of reliability (Fig. A4D, E, F).”
- Change3:
Abstract
Line 10-11: The DeepSurv model showed better performance than the Cox-PH model (C-index: 0.871 vs. 0.866).
- Change4:
Table A2 C-indices of Cox-PH and DeepSurv models based on different hyper-parameters.
Figure A3. ROC curves of the COXCox-PH (A) and DeepSurv (B) models.
Figure A4. Calibration curves of the COXCox-PH (A, B, C) and DeepSurv (D, E, F) models for 3-, 6- and 9-month cancer-specific survival.
Figure A5. Decision-curve analysis of the COXCox-PH (A, B, C) and DeepSurv (D, E, F) models for 3-, 6- and 9-month cancer-specific survival.
Point 2:DeepSurv not only can perform better than COX-PH in general, but also can recommend personalized treatment options. The original article reporting the DeepSurv system emphasizes this aspect in its title: “DeepSurv: personalized treatment recommender system using a Cox proportional hazards deep neural network”. This feature of DeepSurv was mentioned in this manuscript as well in Discussion line 208. This manuscript also reports a section of treatment recommendation, but I cannot find any evidence that this analysis was done by DeepSurv. It appears that the results were an elaboration of the results from COX-PH through K-M plots. If this part was really based on COX-PH, I wonder why it was reported as one of the major findings, worthy to be mentioned in Abstract, Results, and Discussion. By not providing specific details on this analysis, the readers may be misled to the impression that it is based on DeepSurv, as the title emphasizes that this manuscript is about DeepSurv.
Response 2:Thanks a lot for your concern. We are sorry for not giving a clear explanation on the section of treatment recommendation. In fact, the K-M survival analysis was indeed based on the COX-PH, which was mentioned in our manuscript.
(“2. Materials and Methods
2.5. Statistical analysis
Furthermore, Kaplan-Meier (K-M) survival analysis was applied to evaluating the prognostic effect of those independent factors determined by multivariate Cox regression analysis, using log-rank test.”)
This study focused on prognostic analysis in SBTs patients and K-M survival analysis might be helpful for clinical management. Therefore, the results about K-M analysis could be recognized as supplementary explanation of the COX-PH prognostic model.
We are sorry that readers may be misled to the impression that the survival analysis is based on DeepSurv by not providing specific details on this analysis. We have revised the sub-heading of “Treatment Recommendations” and relevant contents. The “Figure A6” was changed as “Figure S1”, as online supplementary material.
- Change1:
Abstract
“Line 8-9: Kaplan-Meier (K-M) survival analysis was performed for further explanation on prognostic effect of Cox-PH model.”
- Change2:
3.Results
“3.4. K-M survival analysis based on the COX-PH model”
- Change3:
Figure legends
“Figure S1. K-M analysis of independent risk factors for cancer-specific survival. A. age; B. prima-ry tumor site (ileum); C. histological type (adenocarcinoma); D, E, F. the efficacy of surgery on prognosis in the sub-groups of patients who had adenocarcinoma (D), had tumor located in ileum (E) and had N2 stage disease (F). (A) age=0 means age < 30, age=1 means 30 < age ≤ 40, age=2 means 40 < age ≤ 50, age=3 means 50 < age ≤ 60, age=4 means 60 < age ≤ 70, age=5 means 70 < age ≤ 80, age=6 means age ≥ 80. (D, E, F) strata=0 means no surgery, strata=1 means surgery, n:1 means Nx, n:2 means N0, n:3 means N1, n:4 means N2.”
- Change4:
4.Discussion
“The penultimate paragraph: In terms of treatment recommendation K-M analysis, our study demonstrated that surgery was a good choice for improving prognosis in patients with SBTs.”
Minor comments:
Point 3:Line 19: “CSS” abbreviation was used in Abstract without full words
Response 3:Thanks for your kind reminder. We have added full words in the Abstract section as followed: “The multivariate analysis demonstrated that seven variables were associated with cancer-specific survival (CSS) (all p<0.05).”
Point 4:Lines 41-45: A brief review of DL on medicine is too broad and brief. It would be better to focus of DL in survival analysis as it is the theme of this manuscript. I think a paragraph in Discussion (lines 204-216) may be a good introduction.
Response 4:Thanks for your kind advice and we have carefully considered the suggestion, making some changes in Discussion as followed: “The remarkable progress in AI facilitates the development of complex nonlinear mod-els based on big clinical data in survival analysis. Adeoye et al. [7] discovered that the DeepSurv model had a higher c-index (0.95 vs 0.83) for assessing the malignant-transformation risk in patients with oral leukoplakia and lichenoid lesions than traditional COX regression analysis. As mentioned in the ‘Introduction’ section, it is indeed practicable to apply DeepSurv algorithm to fitting prognostic models. However, to our knowledge, relative research using DL algorithms for prognostic pre-diction in SBTs are limited. Therefore, this study presents the development of Deep-Surv model for predicting CSS of patients with SBTs and compares the performance of the COX-PH and DeepSurv models.”
Furthermore, the contents in Introduction have also been revised as followed: “Artificial intelligence (AI), consisting of machine learning (ML) and deep learning (DL), has lately been applied to the field of medicine[5-9]. Using an ensemble of ML models, Levi et al.[8] developed a prediction model for detecting patients with gastrointestinal bleeding who required transfusion. Skrede et al.[5] trained a DL model, based on ten-layer convolutional neural networks (CNNs), for risk stratification to help deci-sion-making in the therapy of colorectal cancer. Therefore, AI-aided systems are useful in various clinical setting, e.g., risk stratification, personalized treatments and disease classification based on endoscopic images[10-12]. As for AI application in survival analysis, Katzman et al. [13] first proposed a novel technology, based on deep neural networks, for survival analysis in 2018. They proved that the use of DeepSurv allowed for higher performance of prognostic prediction and effective treatment recommendations, which was superior to the state-of-the-art sur-vival methods such as COX-PH and random survival forests. Since then, DeepSurv has been used in a variety of fields [14-16], e.g., prognosis of lung cancer[17], therapeutic efficacy of chemoradiation in head and neck cancer[18] and detection of cognitive de-cline[19], which all showed better performance than traditional COX-PH models.”
Point 5:In numerous places, “COX regression” was used. It should be replaced to “Cox regression”
Response 5:Many thanks for your concern. We have replaced all “COX regression” with “Cox regression” throughout our manuscript.
Point 6:Line 62: SEER is not “open-source”
Response 6:We apologize for our misnomer. We have deleted the word “open-source” in the manuscript.
“This study was based on the SEER database which provides information on cancer statistics, such as demographic and clinical data, in an effort to re-duce the cancer burden.”
Point 7:Line 104: “cross-validation” was used. Please elaborate on this
Response 7:Thanks for your comment. The cross-validation was mentioned in the original article reporting the DeepSurv system. However, the cross-validation procedure was not used in our training process due to it that both the early-stopping technology and cross-validation procedure were for avoiding overfitting. Therefore, we revised the sentence in our manuscript. (“2. Materials and Methods
2.4. Development of the DeepSurv model
Line 12-14: Early stopping regularization was used to prevent model overtraining and avoid overfitting.”)
Point 8:Table A1: It seems that T stage and N stage were significant due to Tx and Nx. They represent missing data and it is awkward that statistical significance was obtained by missing data.
Response 8:Many thanks for pointing out this issue. Chi-square test was performed between T, N stage and cancer-specific death in the case of removing Tx and Nx using SPSS (V24.0). The results demonstrated that there is no statistical difference in N stage (p=0.731 in the training set, p=0.556 in the validation set), however, there is statistical difference in T stage (p<0.001 in the training set, p=0.001 in the validation set). The table 1 was shown as followed:
Point 9:Table A2: “Katzman et al. [15]” should be “[18]”
Response 9:We apologize for our carelessness. This reference in the Table A2 has been cited using endnote and it can be adjusted in real-time according to the content of our manuscript.
|
Model |
Reference |
C-index |
Layers |
Nodes/Layer |
Dropout |
Learning rate (LR) |
|
Cox-PH |
- |
0.866 |
- |
- |
- |
- |
|
DeepSurv |
Adeoye et al. [7] |
0.863 |
4 |
32, 64, 128, 256 |
0.3 |
0.009545 |
|
0.871 |
4 |
32, 64, 128, 256 |
0.2 |
0.020092 |
||
|
0.849 |
4 |
32, 64, 128, 256 |
0.1 |
0.073907 |
||
|
Katzman et al. [13] |
0.860 |
2 |
48, 48 |
0.147 |
0.089022 |
|
|
0.845 |
3 |
17, 17, 17 |
0.401 |
0.155568 |
||
|
0.838 |
1 |
41 |
0.160 |
0.061359 |
||
|
0.848 |
1 |
45 |
0.109 |
0.061359 |
||
|
0.842 |
1 |
8 |
0.661 |
0.155568 |
Point 10:Lines 178 & 233: The authors claim that N2 stage does not benefit from surgery. But it appears N4 has the adverse effect.
Response 10:Many thanks for your reminding. It is our duty to give explicit interpretations regarding the definition of variables in tables and figures. In the Figure S1F, the definitions N stages are as followed: n:1 = Nx, n:2 = N0, n:3 = N1, n:4 = N2. We have added the explicit explanations in the Figure S1 legend as followed:
“Figure S1. K-M analysis of independent risk factors for cancer-specific survival. A. age; B. primary tumor site (ileum); C. histological type (adenocarcinoma); D, E, F. the efficacy of surgery on prognosis in the sub-groups of patients who had adenocarcinoma (D), had tumor located in ileum (E) and had N2 stage disease (F). (A) age=0 means age < 30, age=1 means 30 < age ≤ 40, age=2 means 40 < age ≤ 50, age=3 means 50 < age ≤ 60, age=4 means 60 < age ≤ 70, age=5 means 70 < age ≤ 80, age=6 means age ≥ 80. (D, E, F) strata=0 means no surgery, strata=1 means surgery, n:1 means Nx, n:2 means N0, n:3 means N1, n:4 means N2.”
Point 11:Figure A6 is illegible. Its quality should be improved. “Age” was categorized to 0-6, without definition.
Response 11:We apologize for our carelessness. We have improved the quality of Figure A6 (Figure S1). Furthermore, the definition of ‘Age’ has been elaborated in figure legend, as mentioned in Response 10.
Point 12:Line 197: “This study” represents yours. It should be “That study”.
Response 12:Thanks for your point. We have revised it in our manuscript as followed: “That study highlighted ten variables, including demographic data (age, marital status) and therapeutic regimen (surgery, chemotherapy).”
Point 13:Lines 253-254 should be removed
Response 13:Many thanks for your careful reminder. We have deleted the sentences in our manuscript. (Line273-274)
Reviewer 2 Report
Manuscript entitled „Development of a Deep Learning Model for Malignant Small Bowel Tumors Survival: a SEER-Based Study” presents an interesting study however, I have some major concerns:
- Precise description of analyzed patients should be included in the material and methods section.
- Boundary and initial conditions should be included in the manuscript.
- Conclusions should be more informative and refer to the presented results.
Author Response
Point 1:Precise description of analyzed patients should be included in the material and methods section.
Response 1:Thanks for your kind reminder. We have added precise description of analyzed patients in the material and methods section, including information on where they came from and the population coverage.
“2. Materials and Methods
2.1. Cohorts dataset
Line 8-11: All patients were selected from the “SEER Incidence Dataset”, which collected cancer incidence data from population-based cancer registries covering nearly 47.9% of the U.S. population, using SEER*Stat software (version 3.8.9.2).”
Point 2:Boundary and initial conditions should be included in the manuscript.
Response 2:Thanks for your advice. Please may I confirm whether it means that the inclusion and exclusion criteria and baseline characteristics should be included? If so, the inclusion and exclusion criteria were described in the “Cohort dataset” section, which might be not well-organized.
(Inclusion criteria: “SBTs were diagnosed using the third edition of International Classification of Disease for Oncology (ICD-O-3), with site codes of C17.0-17.9. For cases diagnosed in 2018 and later, the SEER program applied Extent of Disease (EOD), a set of three items including primary tumor, lymph nodes and metastasis. According to the aforementioned criteria, cases diagnosed with SBTs in 2018 were included in this study.”
Exclusion criteria: “Moreover, the following are the exclusion criteria: (1) patients under the age of 18; (2) patients whose survival duration was unclear; (3) patients with incomplete demo-graphic data.”)
Furthermore, the concise baseline characteristics were described in “Patient Characteristics” section and more precise information were listed in Table A1.
(“3.1. Patient Characteristics
Our study involved a total of 1,987 patients with malignant SBTs, with more males (53.6%) than females (46.4%). A majority of patients with SBTs were between the age of 40 and 80, were white (74.1%), and lived in the metropolis (89.1%) with a comfortable household income. Carcinoid (50.4%) was the most common histological type, followed by adenocarcinoma (30.9%) and stromal sarcoma (11.0%). Primary tumor sites were located in the duodenum (35.1%), ileum (33.3%), jejunum (11.0%) and the overlapping sites (0.8%). Based on the updated American Joint Committee on Cancer (AJCC) 8th edition staging criteria, patients were classified as stage I (24.8%), II (15.5%), III (25.2%), and IV (24.7%). More details were listed in Table A1.”)
Point 3:Conclusions should be more informative and refer to the presented results.
Response 3:Many thanks for your comments. We have revised the conclusions to emphasize the feasibility of DeepSurv model in clinical practices due to the result that DeepSurv performed better than traditional Cox-PH model.
(“In conclusion, our study reports that the DeepSurv model performs better than traditional Cox-PH model in predicting the survival of patients with SBTs. The DL model exhibited feasibility and promise in clinical practices, which might offer insights for future research to explore more DL algorithms in prognostic analysis.”)
Round 2
Reviewer 1 Report
Most of the previous comments have been well addressed. The remaining concerns are as follows:
- In Table A1, the P-values for T-stage and N stage were calculated excluding Tx and Nx as the authors replied. Then it should be mentioned as such in the footnote of Table A1.
- Table A2 reports the result of hyperparameter tuning of DeepSurv along with Cox-Ph. The best models was chosen as the final model, as described in the revised main text. But I am surprised to see the best C-index among hyperparameter tuning, 0.871, was reported in the Abstract. One should report the value from the final model. I suggest the following revision of Table A2: move the hyperparameter tuning results to a supplementary table and show the various statistics of Cox-PH, the internal validation, and the final model. The various statistics may be C-index, 3 AUCs, and Brier index. Lines 193-196 may be revised accordingly.
- Result section 3.4 reports a KM-analysis of Cox-PH and is informative to be further elaborated in Discussion (Lines 269-282). It appears that this result from Cox-PH subgroup analysis can be compared with the personal recommendation analysis by DeepSurv. One may check whether each patient in the validation set is recommended to have surgery or not. Such an analysis would improve the completeness of this manuscript.
Author Response
Point 1:In Table A1, the P-values for T-stage and N stage were calculated excluding Tx and Nx as the authors replied. Then it should be mentioned as such in the footnote of Table A1.
Response 1:Many thanks for your kind advice. We have added the explanation about chi-square tests in T and N stage in the footnote of Table A1.
|   |   | The training set (n=1398) | The validation set (n=589) | ||||
| Characteristics | non-CSD (n=1238) | CSD (n=160) | P | non-CSD (n=534) | CSD (n=55) | P | |
| Sex (%) | 0.404 | 0.320 | |||||
| Female | 566 (45.7) | 67 (41.9) | 258 (48.3) | 31 (56.4) | |||
| Male | 672 (54.3) | 93 (58.1) | 276 (51.7) | 24 (43.6) | |||
| Age (years, %) | <0.001 | <0.001 | |||||
| <30 | 7 (0.6) | 2 (1.2) | 4 (0.7) | 0 (0.0) | |||
| 30-39 | 40 (3.2) | 1 (0.6) | 20 (3.7) | 1 (1.8) | |||
| 40-49 | 100 (8.1) | 2 (1.2) | 42 (7.9) | 1 (1.8) | |||
| 50-59 | 249 (20.1) | 16 (10.0) | 119 (22.3) | 10 (18.2) | |||
| 60-69 | 382 (30.9) | 39 (24.4) | 142 (26.6) | 8 (14.5) | |||
| 70-79 | 308 (24.9) | 53 (33.1) | 143 (26.8) | 14 (25.5) | |||
| ≥80 | 152 (12.3) | 47 (29.4) | 64 (12.0) | 21 (38.2) | |||
| Race (%) | 0.056 | 0.199 | |||||
| Black | 246 (19.9) | 29 (18.1) | 89 (16.7) | 11 (20.0) | |||
| White | 914 (73.8) | 112 (70.0) | 408 (76.4) | 39 (70.9) | |||
| Asian | 70 (5.7) | 18 (11.2) | 33 (6.2) | 3 (5.5) | |||
| Indian | 8 (0.6) | 1 (0.6) | 4 (0.7) | 2 (3.6) | |||
| Median household Income ($, %) | 0.732 | 0.984 | |||||
| (-∞, 35,000) | 19 (1.5) | 1 (0.6) | 10 (1.9) | 1 (1.8) | |||
| [35,000, 55,000) | 271 (21.9) | 38 (23.8) | 106 (19.9) | 12 (21.8) | |||
| [55,000, 75,000) | 539 (43.5) | 66 (41.2) | 222 (41.6) | 23 (41.8) | |||
| [75,000, +∞) | 409 (33.0) | 55 (34.4) | 196 (36.7) | 19 (34.5) | |||
| Region | 0.386 | 0.743 | |||||
| non-metropolis | 141 (11.4) | 14 (8.8) | 55 (10.3) | 7 (12.7) | |||
| metropolis | 1097 (88.6) | 146 (91.2) | 479 (89.7) | 48 (87.3) | |||
| Primary tumor site (%) | <0.001 | <0.001 | |||||
| NOS | 131 (10.6) | 18 (11.2) | 51 (9.6) | 5 (9.1) | |||
| Duodenum | 537 (43.4) | 102 (63.7) | 210 (39.3) | 38 (69.1) | |||
| Jejunum | 145 (11.7) | 15 (9.4) | 57 (10.7) | 2 (3.6) | |||
| Ileum | 419 (33.8) | 23 (14.4) | 210 (39.3) | 9 (16.4) | |||
| Overlapping | 6 (0.5) | 2 (1.2) | 6 (1.1) | 1 (1.8) | |||
| Histological type (%) | <0.001 | <0.001 | |||||
| others | 72 (5.8) | 31 (19.4) | 40 (7.5) | 10 (18.2) | |||
| Adenocarcinoma | 320 (25.8) | 111 (69.4) | 143 (26.8) | 39 (70.9) | |||
| Carcinoid | 685 (55.3) | 16 (10.0) | 295 (55.2) | 6 (10.9) | |||
| Stromal sarcoma | 161 (13.0) | 2 (1.2) | 56 (10.5) | 0 (0.0) | |||
| T stage (%) | <0.001 | 0.001 | |||||
| Tx | 164 (13.2) | 68 (42.5) | 77 (14.4) | 23 (41.8) | |||
| T0 | 3 (0.2) | 2 (1.2) | 0 | 0 | |||
| Tis | 0 (0.0) | 1 (0.6) | 0 | 0 | |||
| T1 | 298 (24.1) | 22 (13.8) | 112 (21.0) | 9 (16.4) | |||
| T2 | 176 (14.2) | 5 (3.1) | 84 (15.7) | 1 (1.8) | |||
| T3 | 340 (27.5) | 17 (10.6) | 146 (27.3) | 5 (9.1) | |||
| T4 | 257 (20.8) | 45 (28.1) | 115 (21.5) | 17 (30.9) | |||
| N stage (%) | 0.731 | 0.556 | |||||
| Nx | 109 (8.8) | 37 (23.1) | 53 (9.9) | 12 (21.8) | |||
| N0 | 687 (55.5) | 79 (49.4) | 260 (48.7) | 27 (49.1) | |||
| N1 | 303 (24.5) | 29 (18.1) | 150 (28.1) | 11 (20.0) | |||
| N2 | 139 (11.2) | 15 (9.4) | 71 (13.3) | 5 (9.1) | |||
| M stage (%) | <0.001 | <0.001 | |||||
| M1 | 983 (79.4) | 78 (48.8) | 408 (76.4) | 27 (49.1) | |||
| M2 | 255 (20.6) | 82 (51.2) | 126 (23.6) | 28 (50.9) | |||
| Stage (%) | <0.001 | <0.001 | |||||
| NA | 117 (9.5) | 27 (16.9) | 42 (7.9) | 9 (16.4) | |||
| I | 342 (27.6) | 16 (10.0) | 132 (24.7) | 3 (5.5) | |||
| II | 207 (16.7) | 17 (10.6) | 77 (14.4) | 6 (10.9) | |||
| III | 317 (25.6) | 18 (11.2) | 157 (29.4) | 9 (16.4) | |||
| IV | 255 (20.6) | 82 (51.2) | 126 (23.6) | 28 (50.9) | |||
| Surgery (%) | <0.001 | <0.001 | |||||
| No | 288 (23.3) | 114 (71.2) | 123 (23.0) | 40 (72.7) | |||
| Yes | 950 (76.7) | 46 (28.7) | 411 (77.0) | 15 (27.3) | |||
| Tumor size (mm, median [IQR]) |
21.00 [12.00, 42.00] | 32.00 [20.00, 55.00] | <0.001 | 22.00 [12.00, 41.00] | 40.00 [25.50, 53.00] | <0.001 | |
| Removed nodes (median [IQR]) |
3.00 [0.00, 15.00] | 0.00 [0.00, 1.00] | <0.001 | 3.00 [0.00, 15.00] | 0.00 [0.00, 0.00] | <0.001 | |
| Lymph node examine (%) | <0.001 | <0.001 | |||||
| No | 522 (42.2) | 115 (71.9) | 222 (41.6) | 41 (74.5) | |||
| Yes | 716 (57.8) | 45 (28.1) |   | 312 (58.4) | 14 (25.5) |   | |
| CSD=cancer-specific death; NOS=no other specific; NA=not available. The P-values for T-stage and N stage were calculated excluding Tx and Nx. | |||||||
Point2:Table A2 reports the result of hyperparameter tuning of DeepSurv along with Cox-Ph. The best model was chosen as the final model, as described in the revised main text. But I am surprised to see the best C-index among hyperparameter tuning, 0.871, was reported in the Abstract. One should report the value from the final model. I suggest the following revision of Table A2: move the hyperparameter tuning results to a supplementary table and show the various statistics of Cox-PH, the internal validation, and the final model. The various statistics may be C-index, 3 AUCs, and Brier index. Lines 193-196 may be revised accordingly.
Response 2:Thanks a lot for your concern. We have move the hyperparameter tuning results to a supplementary table as Table S1.
Furthermore, the performance of the Cox-PH and DeepSurv models was listed in the Table A2 as followed:
Table A2. Performance of the COX-PH and DeepSurv models.
|
Model |
Group |
C-index |
AUCs |
Brier scores |
||||
|
3-month |
6-month |
9-month |
3-month |
6-month |
9-month |
|||
|
Cox-PH |
The training set (n=1398) |
0.878 |
0.898 |
0.897 |
0.894 |
0.262 |
0.252 |
0.246 |
|
The validation set (n=589) |
0.866 |
0.874 |
0.922 |
0.908 |
0.282 |
0.265 |
0.265 |
|
|
DeepSurv |
The internal-validation set (n=269) |
0.869 |
0.874 |
0.871 |
0.871 |
0.068 |
0.079 |
0.085 |
|
The validation set (n=589) |
0.871 |
0.878 |
0.891 |
0.891 |
0.058 |
0.070 |
0.080 |
|
Additionally, we revised the manuscript. 1) Line 169-171: Furthermore, the C-index, AUCs and Brier scores based on the training set were demonstrated in Table A2. 2) Line 208-209: The performance of the DeepSurv model was listed in Table A2 based on the internal-validation set.
Point3:Result section 3.4 reports a KM-analysis of Cox-PH and is informative to be further elaborated in Discussion (Lines 269-282). It appears that this result from Cox-PH subgroup analysis can be compared with the personal recommendation analysis by DeepSurv. One may check whether each patient in the validation set is recommended to have surgery or not. Such an analysis would improve the completeness of this manuscript.
Response 3:Many thanks for your kind reminder and we agree with your point. We are fully aware of the importance of the personal recommendation analysis by DeepSurv. However, this study focused on the impact of models on the prognosis rather than treatment recommendation. Furthermore, we will focus attention on this analysis in future research.

Reviewer 2 Report
The authors answered my questions and corrected the manuscript.
Author Response
Point 1:The authors answered my questions and corrected the manuscript.
Response 1:Thanks a lot for your constructive comments on our manuscript. We believe that our manuscript has got greatly improved by addressing these comments.

Round 3
Reviewer 1 Report
There is one outstanding minor revision that should be made.
The section 3.3. reports the performance of DeepSurv, which the authors have tried to revise together with Table A2, following my previous comments. But inconsistency was noticed as follows:
The hyperparameter tuning results have moved to Table S1, as I suggested. The best C-index there was 0.871, but the revised Table A2 shows C-index for the internal validation set as 0.869. The hyperparameter tuning must have been done with the internal validation set. How can these two numbers be different?
Furthermore, Line 194 mixed the result of the validation set (C-index of 0.871) with the result of hyperparameter tuning: it can be confusing. Lines 192~202 should be revised as follows:
"rameters. We tuned DeepSurv's hyper-parameters ... as described in Table S1. With the best performing hyper-parameters, four layers (nodes: ... of 0.020092, the performance of the final model was evaluated with the validation set (n=589). The C-index was 0.871(?), while the AUCs were 0.878, ... as shown in Table A2 and Fig. A3B. Furthermore, ... (Fig. A4D, E, F). DCA indicated that this model had similar"
Author Response
Point 1:The section 3.3. reports the performance of DeepSurv, which the authors have tried to revise together with Table A2, following my previous comments. But inconsistency was noticed as follows:
The hyperparameter tuning results have moved to Table S1, as I suggested. The best C-index there was 0.871, but the revised Table A2 shows C-index for the internal validation set as 0.869. The hyperparameter tuning must have been done with the internal validation set. How can these two numbers be different?
Response 1:Many thanks for your concern. The revised Table A2 demonstrated that the C-index of DeepSurv model is 0.869 in the internal-validation set (n=269) and 0.871 in the validation set (n=589). The performance of DeepSurv model is improved according to hyperparameter tuning in the internal validation set (n=589) and the best C-index is 0.869. With this best performing hyperparameters, the C-index is 0.871 in the validation set (n=589). Therefore, these two numbers are different.
Point2:Furthermore, Line 194 mixed the result of the validation set (C-index of 0.871) with the result of hyperparameter tuning: it can be confusing. Lines 192~202 should be revised as follows:
"rameters. We tuned DeepSurv's hyper-parameters ... as described in Table S1. With the best performing hyper-parameters, four layers (nodes: ... of 0.020092, the performance of the final model was evaluated with the validation set (n=589). The C-index was 0.871(?), while the AUCs were 0.878, ... as shown in Table A2 and Fig. A3B. Furthermore, ... (Fig. A4D, E, F). DCA indicated that this model had similar".
Response 2:Many thanks for your kind advice. We have revised this paragraph according to your comments and it is indeed greatly improved and clearly organized.
“The aforementioned training set (n=1398) was randomly split into the “internal training” set (n=1129) and the “internal validation” set (n=269) at 8:2 for the best set of hyperparameters. The validation set (n=589) was regarded as the “testing” set for evaluating the performance of the final DeepSurv model. We tuned DeepSurv’s hyperparameters by referring to previous studies[7, 13], as described in Table S1. With the best performing hyperparameters, four layers (nodes: 32, 64, 128, 256), dropout rate of 0.2 and learning rate of 0.020092, the performance of the final model was evaluated with the validation set (n=589). The C-index was 0.871, while the AUCs were 0.878, 0.891 and 0.891 for 3-, 6- and 9-month CSS, respectively, which were inferior to those of the Cox-PH model, as shown in Table A2 and Fig. A3B. Furthermore, the calibration and prediction curves of the DeepSurv model for the 3-, 6- and 9-month CSS showed great correlations in the validation cohort, with brier scores of 0.058, 0.070 and 0.080, respectively, indicating a high degree of reliability (Fig. A4D, E, F). DCA indicated that this model had similar net benefits as the Cox-PH model, demonstrating both of them owned good clinical practice in predicting CSS in patients with SBTs (Fig. A5D, E, F).”
